DOI: 10.1038/s41467-018-05534-5　　**OPEN**

# Creating solvation environments in heterogeneous catalysts for efficient biomass conversion

Qi Sun[1], Sai Wang[1], Briana Aguila [2], Xiangju Meng[1], Shengqian Ma [2] & Feng-Shou Xiao[1]

Chemical transformations are highly sensitive toward changes in the solvation environment and solvents have long been used to control their outcome. Reactions display unique performance in solvents like ionic liquids or DMSO, however, isolating products from them is cumbersome and energy-consuming. Here, we develop promising alternatives by constructing solvent moieties into porous materials, which in turn serve as platforms for introducing catalytic species. Due to the high density of the solvent moieties, these porous solid solvents (PSSs) retain solvation ability, which greatly influences the performance of incorporated active sites via concerted non-covalent substrate–catalyst interactions. As a proof-of-concept, the -SO$_3$H-incorporated PSSs exhibit high yields of fructose to 5-hydroxymethylfurfural in THF, which exceeds the best results reported using readily separable solvents and even rivals those in ionic liquids or DMSO. Given the wide application, our strategy provides a step forward towards sustainable synthesis by eliminating the concerns with separation unfriendly solvents.

[1] Key Lab of Applied Chemistry of Zhejiang Province and Department of Chemistry, Zhejiang University, Hangzhou 310028, China. [2] Department of Chemistry, University of South Florida, 4202 E. Fowler Avenue, Tampa, FL 33620, USA. These authors contributed equally: Qi Sun, Sai Wang. Correspondence and requests for materials should be addressed to Q.S. (email: sunqichs@zju.edu.cn) or to F.-S.X. (email: fsxiao@zju.edu.cn)

Catalytic systems that are active and selective toward organic transformations with readily separable products are highly sought after, while optimizing them is a multifaceted process that is as frustrating as it is compulsory[1–3]. One strategy to enhance the performance of abiological catalysts is to engage the reaction participants with specific non-covalent interactions to perturb the energies of the competing transition states and thereby the overall efficiency, in a way reminiscent of enzymatic catalysis[4–8]. To harness non-covalent interactions to exert control over the reactions, solvent screenings are preferential and varying the reaction media often results in great divergent outcomes in terms of both activity and selectivity[9–13]. A wide range of reactions exhibits unique performance, being operated in polar aprotic solvents[14–16], such as dimethyl sulfoxide (DMSO), 1-methyl-2-pyrrolidinone (NMP), and ionic liquids. However, the irreplaceable role of these solvents in organic synthesis is contrasted with their extreme complicacy in separation of the products, and even worse, side reactions are often accompanied with the separation process[17–20]. To counter these concerns, the development of alternatives that could mimic the solvation environment of these solvents to aid the accomplishment of those transformations, is highly desirable.

In nature, rather complicated organic compounds are often constructed elegantly in cell via catalysis in flawlessly engineered spaces, with catalytically active sites surrounded by amino acid residues to be complementary to the transition states of the reactions via concerted non-covalent substrate–catalyst interactions[21,22]. Inspired by nature, we envisioned that providing a suitable reaction environment that mimics those solvents in close proximity of the active sites in heterogeneous catalysts and performing the reaction using a separate-friendly solvent instead, for mass and heat transfer, may offer prospective solutions to the aforementioned challenges pertaining to the separation of the products from high boiling point solvents or nonvolatile ionic liquids. However, it remains challenging to transfer such complex systems from liquid phase to the realm of solid state[23], because this requires precise control of the spatial continuity and dynamic interactions between the immobilized partners. To fulfill this task, a microenvironment that possesses solvent-like behavior together with the following attributes is highly desired: (I) amenable synthesis to enable the incorporation of various catalytic components for engineering the chemical microenvironment around the active sites; (II) adjustable chemical composition, allowing for careful positioning of individual functions and forming a well-defined array of non-covalent interactions around a catalytic site thus providing ways for improving performance by tailoring synergistic interactions; (III) high surface area with hierarchical porosity to enable fast mass transfer; and (IV) robustness under various chemical conditions, allowing for long-term stable performance and potential recycling to simplify the workup procedures. Such a microenvironment can be targeted by constructing judiciously designed reaction participants into porous organic

polymers with the following considerations: the flexibility and modularity of polymer chains enable the provision of a solvent-like reaction environment, onto which catalytically active sites can be grafted in a predefined way that can be tuned with precision at the molecular scale. In addition to these, the tunable pore structures and exceptional chemical stability of porous polymers further make them significantly attractive[24–33] (Fig. 1).

To implement this strategy, we first modify the solvent moieties with specific functionality for the potential construction of porous frameworks. Earlier work has established that porous polymers can be generated by the solvothermal polymerization of vinyl-functionalized monomers with the advantages of excellent functional group compatibility, adjustable composition, high yield, and tunable pore structures[34,35]. With respect to the active sites, a diverse range of catalytic moieties is amenable for this role, for example acid or base, or groups that coordinate metal ions. Here, given the versatility of the sulfonic acid group[36–41], it is the catalytically active site of choice for demonstrating this proof-of-concept study. We anticipated that, embedding the acid groups into polymeric analogs of desirable solvents, via noncovalent interactions with the reaction participants, the surrounded solvent mimic environment may be able to affect the acid catalyzed reaction rate or alter the selectivity of product formation. Indeed, in the dehydration of fructose to produce 5-hydroxymethylfurfural (HMF), an important transformation from biomass to valuable chemicals that performed well in solvents, such as DMSO and ionic liquids, we obtain the highest reported HMF yields to date in a monophasic, readily separable solvent, such as THF, competitive to those using corresponding monomeric solvent analogs as reaction media. Our results therefore provide a step forward toward achieving a sustainable biomass-based chemicals and fuels platform by greatly simplifying the procedure for the target product. Given the compositional tunability of porous solid solvents, this strategy may herald the advent of a new avenue to design efficient and green chemical processes.

## Results

**Synthesis of porous solid solvents (PSSs).** Our initial step was to construct monomeric solvent analogs into highly porous frameworks. Various solvent moieties (NMP, DMSO, and imidazolium-type ionic liquid,) were functionalized with styryl to afford V-NMP, V-DMSO, and V-IL, respectively (Table 1). The construction of these monomers into porous polymers was achieved by polymerization of these monomers in dimethylformamide (DMF) at 100 °C for 24 h in an autoclave with the assistance of free radical initiator of azobisisobutyronitrile (AIBN), according to the polymerization method developed by our group[42]. It is noteworthy that this process gives rise to the polymers in quantitative yields and these vinyl-functionalized monomeric solvent analogs can be readily obtained from

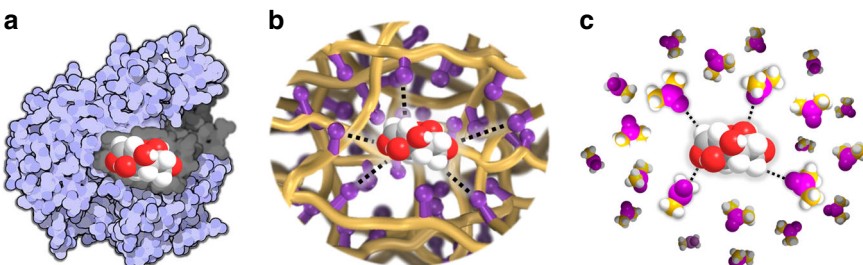

**Fig. 1** Schematic illustration of reactant as representative by fructose in various systems. **a** A flawlessly engineered space of enzyme, **b** a microenvironment with solvent-like behavior inspired by nature and solution system, and **c** solvent containing hydrogen bond acceptors

commercially available reagents, with details provided in the Supplementary Information (see details in the experimental section). Due to the flexibility of the polymer chains, the resultant polymers can be well dispersed in various solvents, suggesting that they can provide solvent-like environments (Supplementary Fig. 1). However, the insoluble nature allows them to be readily separable from various reaction systems. These easy-to-achieve and separation-friendly properties give them great promise for practical applications.

**Physiochemical characterization and local structure analysis.** As a representative sample among the synthesized porous solid solvents (PSSs), the PSS bearing imidazolium-type ionic liquid moieties (PSS-IL) is illustrated thoroughly. To examine the local chemical composition of PSS-IL, $^{13}C$ MAS NMR analysis was

performed (Fig. 2a). The successful transformation from the vinyl-functionalized imidazolium-type ionic liquid into highly polymerized material is verified by the disappearance of peaks in the range of 110.0–120.0 ppm related to vinyl groups and the concomitant emergence of a strong peak at 40.7 and 59.0 ppm attributable to the polymerized vinyl groups form styrene and 1-vinylimidazole moieties, respectively (details of peaks assignment see Supplementary Fig. 2)[43,44]. The morphology of PSS-IL was studied by scanning electron microscopy (SEM) and transmission electron microscopy (TEM), showing that PSS-IL displayed a rough surface that is composed of randomly agglomerated small particles with sizes ranging from several nanometers to over tens of nanometers, yielding interconnected meso- and macroporous ensembles (Fig. 2b, c). To investigate the details of the pore properties of PSS-IL, $N_2$ sorption isotherms at −196 °C were collected (Fig. 2d). It is shown that PSS-IL exhibits type-I plus type-IV sorption curves, whereby a steep step in the curve at relative pressure ($P/P_0$) <0.01 is due to the filling of micropores, while a hysteresis loop at $P/P_0$ in the range of 0.5–0.95 is mainly from the contribution of the sample mesoporosity. Analysis of pore-size distribution evaluated by the nonlocal density functional theory (NLDFT) indicates that its pore sizes were predominantly distributed around 0.5 and 50 nm (Supplementary Fig. 3), confirming its hierarchical porosity. Derived from the $N_2$ adsorption data, the Brunauer–Emmett–Teller (BET) surface area and pore volume of PSS-IL were calculated to be 460 $m^2\,g^{-1}$ and 0.60 $cm^3\,g^{-1}$, respectively. The details of the characterization of PSS-NMP and PSS-DMSO are provided in the Supplementary Information (see Supplementary Fig. 4–11). It is rather remarkable that the PSSs exhibit excellent hydrothermal stability as demonstrated by the fact that after 1 week of boiling water treatment, negligible decreases in surface area and pore volume were observed (Supplementary Table 1).

**Solvation effect.** Solvation describes the interaction of solvents with molecules or ions in a solute. The primary functions of

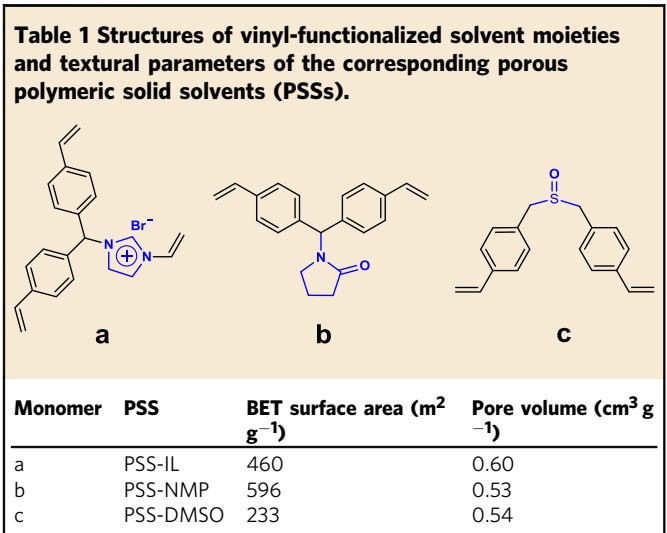

**Table 1 Structures of vinyl-functionalized solvent moieties and textural parameters of the corresponding porous polymeric solid solvents (PSSs).**

| Monomer | PSS | BET surface area ($m^2\,g^{-1}$) | Pore volume ($cm^3\,g^{-1}$) |
|---|---|---|---|
| a | PSS-IL | 460 | 0.60 |
| b | PSS-NMP | 596 | 0.53 |
| c | PSS-DMSO | 233 | 0.54 |

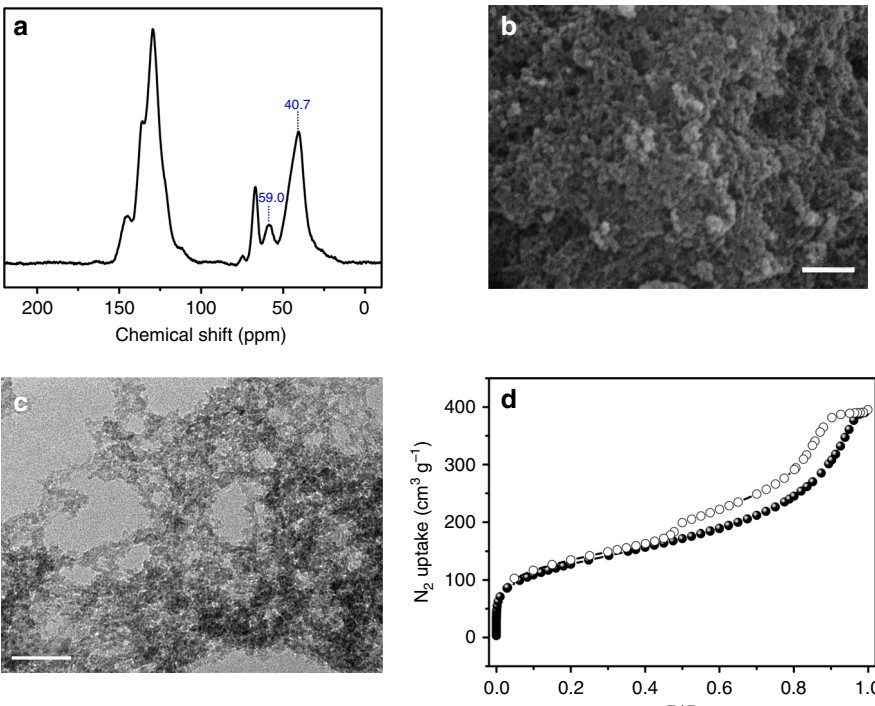

**Fig. 2** Characterizations of PSS-IL. **a** $^{13}C$ MAS NMR spectrum, **b** SEM image (scale bar denotes 500 nm), **c** TEM image (scale bar denotes 100 nm), and **d** $N_2$ sorption isotherms collected at −196 °C of PSS-IL

solvents in chemical syntheses, such as dissolving the reagents for facilitating mass transfer and modulating chemical reactions in terms of reaction activity and selectivity are all a result of solvation. For example, it has been well known that numerous transformations in relation to carbohydrates exhibit excellent solubility and unique performance in the presence of solvents like ionic liquids (ILs), DMSO, and NMP, due to the strong solvation effect between these solvents and the carbohydrates[45–48]. The solvation mechanism lying between those solvents and the carbohydrates is proposed to involve in the interaction of the hydrogen bond acceptors, such as halide ions in ILs, S=O in DMSO, or C=O in NMP, with the hydroxyl protons of the carbohydrates to break the extensive hydrogen bonding networks, thereby promoting their dissolution as well as perturbing the reaction. Indeed, theoretical calculations and experimental evidences demonstrate that in a great number of carbohydrate involved transformations, some key intermediates identified in the reaction pathway are a complex that incorporates with those solvents to stabilize them, thus favoring the subsequent desired transformation[49].

Given the excellent spatial continuity of these solvent moieties in the three dimensional nanospace of PSSs, we therefore suggest that similar hydrogen bonding interactions should also be considered in their solid state and the densely populated hydrogen bond acceptors in the PSSs are supposed to be effective in disrupting and breaking the intramolecular hydrogen-bonding network present in the carbohydrates, thus leading to the decreased crystallinity. To test this hypothesis, we mixed the PSSs with fructose, a typical carbohydrate, at a mass ratio of 10:1, respectively, and mechanically grinded to facilitate their interaction. To exclude the role of the porous polymer backbone and to highlight the unique property of those PSSs, a porous polymer without a hydrogen bond acceptor, polydivinylbenzene (PDVB), was synthesized for comparison (Supplementary Fig. 12). Due to the crystalline nature of fructose, it exhibits strong XRD diffraction peaks ranging from 10 to 40° (Supplementary Fig. 13). It is very interesting to find that after grinding fructose with PSSs, the XRD peaks associated with fructose disappeared completely in the resultant composites, while these peaks were retained in the composite of PDVB and fructose, and prolonging the grinding time had negligible effect on the crystallinity of fructose as evidenced by the retained intensity of the XRD peaks related to fructose (Supplementary Fig. 14). These results suggest that porous polymers constructed by monomers containing a hydrogen bond acceptor can effectively break the hydrogen network of fructose via coordinating the fructose hydroxyl groups to the hydrogen bond acceptor groups in PSSs like their liquid analogs, exhibiting a solvation effect to fructose. To provide additional proof, Fourier transform infrared (FT-IR) analysis was carried out. It was shown that the C=O vibration shifted from 1677 cm$^{-1}$ for pristine PSS-NMP to 1663 cm$^{-1}$ for the composite of finely grinded PSS-NMP and fructose (fructose@PSS-NMP), and meanwhile the peak associated with the OH group of fructose shifted from 1044 to 1054 cm$^{-1}$ of the composite, thereby confirming that strong interactions exist between fructose and the NMP moieties in PSS-NMP (Supplementary Fig. 15). To provide more insights about such interactions, variable temperature IR spectra were collected. Negligible changes in the IR spectra of fructose@PSS-NMP along with the temperature increase were observed, even after heating up to 100 °C, suggestive of the strong binding (Supplementary Fig. 16).

**Introduction of active species into PSSs.** After proving the solvation ability of PSSs, we proceeded to introduce catalytic active species and the sulfonic acid group was chosen. Taking

PSS-IL as an example, a family of acid catalysts with different active site concentrations were readily obtained via co-polymerization of V-IL and sodium $p$-styrene sulfonate with various ratios (PSS-xIL-SO$_3$Na), followed by ion-exchange with 1 M HCl (PSS-xIL-SO$_3$H), where $x$ represents the molar ratio of V-IL and sodium $p$-styrene sulfonate. The successful incorporation of sulfonic acid functionality was confirmed by X-ray photo-electron spectroscopy (XPS), FT-IR, energy dispersive X-ray (EDX) mapping, and elemental analysis. The XPS spectra of PSS-xIL-SO$_3$Na revealed the sulfur and sodium signals at binding energies of 163 and 1070 eV, respectively, suggestive of the presence of sodium sulfonate species (Supplementary Fig. 17). In addition, compared with PSS-IL, PSS-xIL-SO$_3$Na samples show additional bands at ca. 1010, 1126, 1185, and 1033 cm$^{-1}$, which are assigned to O=S=O and C-S bond, characteristics of the sulfonate species (Supplementary Fig. 18)[50]. EDX analysis in a transmission electron microscope (TEM) provided evidence that sulfur and sodium elements, signatures of –SO$_3$Na groups, are indeed located throughout the resultant copolymers (Supplementary Fig. 19 and 20), indicative of their homogeneous distribution. Before catalytic evaluation, PSS-xIL-SO$_3$Na polymers were immersed into an HCl solution (1 M) to replace Na$^+$ with H$^+$. The complete cation-exchange process was confirmed by TEM-EDS elemental mapping and XPS analyses (Supplementary Fig. 17, 19, and 20), as evidenced by the fact that negligible sodium species were detected in PSS-xIL-SO$_3$H samples. To determine the content of sulfonic acid groups in PSS-xIL-SO$_3$H, infrared absorption carbon–sulfur analysis and acid-base titration experiments were carried out, showing that the experimental S content was consistent with the theoretical values (Supplementary Table 2). Moreover, to evaluate the acid strength of the sulfonic groups in the resultant samples, studies associated with the adsorption of a probe molecule were carried out. Given the sensitivity and isotropicity of $^{31}$P chemical shift of the probe trimethylphosphine oxide (TMPO) according to its interaction strength with a Brønsted acid site, it has proven to be an informative tool for identifying the acidity of multiple acid sites and thereby is our choice. $^{31}$P MAS NMR spectra of TMPO after interaction with PSS-30IL-SO$_3$H and PSS-5IL-SO$_3$H samples shows a singlet peak at 64.8 and 65.3 ppm, respectively, appearing to possess a moderate strength. The narrow NMR signal suggests the homogeneity of the acid sites in these samples (Supplementary Fig. 21)[51,52]. To prove the accessibility of the active sites by reagents, N$_2$ sorption isotherms were collected at −196 °C (Supplementary Fig. 22). The textural parameters derived from the sorption isotherms are summarized in Supplementary Table 3, indicating that they also exhibit high BET surface areas (367–500 m$^2$ g$^{-1}$), large pore volumes (0.60–0.81 cm$^3$ g$^{-1}$), and hierarchical porosities.

**Evaluation of catalytic performance.** To test the efficiency of the resultant catalysts, we set out to evaluate their performance in the selective dehydration of fructose to produce 5-hydroxymethylfurfural (HMF). It is also based on the following considerations:[53–59] (1) HMF is a very important compound serving as a building block platform bridging biomass chemistry and petro chemistry; (2) this transformation is highly solvent dependent, which performs well in the presence of solvents like DMSO, NMP, or ionic liquids, however, it remains a substantial challenge to separate HMF in an energy-friendly manner from those solvents. To demonstrate the possibility of PSSs as an alternative to the corresponding solvent in regulating the performance of the acid sites, initial catalytic evaluations were conducted using readily separable THF as a reaction medium. A set of control experiments were conducted to illustrate the

**Table 2 Catalytic data in the dehydration of fructose to HMF over various catalysts using THF as a solvent.[a]**

| Entry | Catalyst | Time (min) | Conv.(%) | Select.(%) | Yield (%) |
|---|---|---|---|---|---|
| 1 | PSS-30IL-SO$_3$H | 10 | >99.5 | 98.8 | 98.8 |
| 2 | Amberlyst-15 | 120 (10) | 61.7 (5.4) | 21.2 (70.4) | 13.1 (3.8) |
| 3 | TsOH | 120 (10) | 95.1 (11.4) | 29.1 (76.3) | 26.6 (8.7) |
| 4[b] | TsOH | 120 | >99.5 | 64.5 | 64.5 |
| 5[b] | Amberlyst-15 | 120 | 58.4 | 24.3 | 14.2 |
| 6[c] | TsOH | 120 (10) | >99.5 (43.2) | 72.8 (89.3) | 72.8 (38.6) |
| 7[c] | Amberlyst-15 | 120 (10) | >99.5 (28.7) | 64.1 (82.8) | 64.1 (23.8) |
| 8[d] | PSS-30IL-SO$_3$H | 10 | >99.5 | 98.7 | 98.7 |
| 9[e] | PSS-30IL-SO$_3$H | 10 | >99.5 | 97.8 | 97.8 |

[a]Reaction conditions: fructose (100 mg, 0.56 mmol), catalyst (based on the amount of H$^+$ 1.0 mol%), 120 °C, THF (5.0 mL)
[b]Addition of 100 mg of PSS-IL
[c]Addition of 1-ethyl-3-methyl imidazolium bromide (34 mg, containing same mole amount of ionic moiety to that in PSS-30IL-SO$_3$H)
[d]Fructose (2.0 g), PSS-30IL-SO$_3$H (1.0 mol%), 120 °C, and THF (40 mL) for 10 min
[e]Recycle for five times. The values in parentheses refer to the time used, as well as the conversion of fructose and selectivity and yield of HMF at that point

advantages of the PSSs-SO$_3$H catalytic systems. As presented in Table 2, PSS-30IL-SO$_3$H shows exceptional catalytic activity in the dehydration of fructose to form HMF, outcompeting all other catalytic systems tested under identical conditions. A full fructose conversion and a HMF yield of 98.8% were achieved for PSS-30IL-SO$_3$H within 10 min, which compares far more favorably to the corresponding values of two benchmark catalysts Amberlyst-15 (Table 2, entry 2) and 4-methylbenzenesulfonic acid (TsOH, Table 2, entry 3), affording fructose conversions of 5.4% and 11.4% as well as HMF yields of 3.8% and 8.7%, respectively, at that time. Considering that these catalysts bear the same type of acid species and the catalytic inactivity of PSS-IL (Supplementary Table 4), we therefore infer that the synergetic effects of sulfonic acid groups and densely populated ionic moieties are responsible for the dramatic increase in HMF yield of PSS-30IL-SO$_3$H. This is further supported by the fact that PSS-IL can greatly improve the performance of TsOH in terms of selectivity by a factor of about 2.5, while it has negligible effect on the performance of Amberlyst-15 (Table 2, entries 4 and 5). In the last scenario, the acid sites and ionic moieties are spatially isolated, as they are situated on separate solid materials, thereby leading to their incompetent cooperation. On the contrary, the homogeneous catalyst TsOH is easily diffused to the nearby ionic moieties in PSS-IL, allowing for cooperation.

To further support the cooperative effect between the ionic moieties and the acid groups, a monomeric analog of PSS-IL, 1-ethyl-3-methyl imidazolium bromide (EMIMBr), was introduced into Amberlyst-15 and TsOH catalytic systems. Enhanced activities were observed for both of the catalytic systems, giving rise to fructose conversions of 28.7% and 43.2% as well as HMF yields of 23.8% and 38.6%, respectively, after 10 min (Table 2, entries 6 and 7); nonetheless, they are still much lower than the yield obtained by PSS-30IL-SO$_3$H (98.8%). This observation can be explained as follows: the free mobility of the molecular ionic compounds enables their cooperation with catalytic sites in both solid and liquid states, thereby promoting the reaction. On the other hand, as the molecular compound homogeneously distributes in the solution, a portion of them are dissolved in the solvent and do not participate in the cooperative catalytic reaction, thus resulting in the relatively lower utilization efficiency. In contrast, all of the ionic moieties in PSS-IL are enriched in the vicinity of the acid species in the confined space,

which therefore dramatically facilitate their cooperation, leading to exceedingly high activity. This observation suggests the superiority of our design by constructing a desired solvent-like environment around the active sites in solid materials, which not only enables easy separation but also increases their utilization efficiency.

**Importance of a high density of solvent moieties in the catalysts.** The results above imply that the density of the ionic moieties has a great effect on the performance of the acid sites. In order to gain better insight into this idea, the catalytic performances of catalysts with different molar ratios of V-IL and sulfonic acid were evaluated. Assessment of the catalytic responses of these materials in the dehydration of fructose to form HMF was displayed in Fig. 3a (see also Supplementary Table 5). Upon an increase of the molar ratio of V-IL with sulfonic acid, augmented productivity as well as selectivity to HMF was observed. In line with its highest ionic component content among the samples tested, PSS-30IL-SO$_3$H exhibited the highest HMF yield (98.8%), surpassing all of the state-of-the-art materials tested as well as approaching or exceeding the reaction performed using ionic liquid or DMSO as a reaction medium (Supplementary Tables 5 and 6). As the mole ratio decreased from 30 to 5, the HMF yield decreased substantially from 98.8% to 13.6%. To facilitate the comparison with other catalytic systems, TOF values were calculated. To keep the fructose conversion below 15% and thereby ensuring the accuracy of the TOF values calculation, catalytic reactions were performed under a relatively low catalyst loading (0.25 mol% of H$^+$) in the presence of THF at 120 °C. PSS-xIL-SO$_3$H ($x = 30$, 20, 15, 10, 5) afforded initial reaction rates of 0.064, 0.048, 0.042, 0.032, and 0.0095 mM min$^{-1}$, respectively, at the beginning 5 min. These results imply the importance of the density of the ionic component on the catalytic performance of the resultant catalysts. Possibly, a high concentration of ionic moieties is necessary to effectively mimic its solvent analog to modulate the reaction. To rationalize this assumption, the ability of various samples with different densities of ionic moieties for breaking of the fructose hydrogen network was evaluated. To avoid the interference of the acid groups, a non-polar molecule, divinylbenzene (DVB), was used to adjust the density of the ionic moieties in the resultant polymers (Supplementary Table 7). As expected, the V-IL/DVB mole ratio was found to be important for

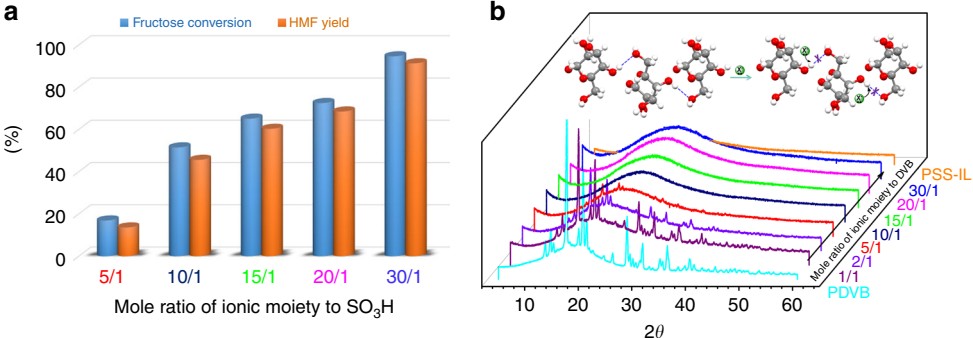

**Fig. 3** The dependence of catalytic performance on the solvation ability of the catalysts. **a** The effect of mole ratio of ionic moieties to -SO₃H groups on the catalytic performance of dehydration of fructose to form HMF. Reaction conditions: fructose (100 mg), catalyst (1.0 mol%), THF (5.0 mL), and 120 °C for 10 min. **b** XRD patterns of finely grinded polymers (PSS-xIL-DVB, x stands for the mole ratio of IL groups to divinylbenzene, 100 mg) and fructose (10 mg). Inset: schematic illustration of the interaction of the hydrogen bond acceptors, halide ions in ionic moieties, with the hydroxyl protons of the carbohydrates to break the extensive hydrogen bonding networks

the overall solvation efficiency of the polymer toward fructose (Fig. 3b). When the V-IL/DVB mole ratio is lower than 2, there is crystalline fructose in the finely grinded composite. As the V-IL/DVB mole ratio increased to higher than 10, the peaks related to the fructose crystallinity disappear completely, indicating that the crystalline fructose is fully transformed into an amorphous phase as a result of solvation by the polymers.

**Stability and recyclability investigation**. Given the excellent performance, PSS-30IL-SO₃H was chosen as a representative sample for further investigation. To further evaluate its catalytic efficiency, we screened a range of temperatures and solvent systems to monitor the HMF formation. It is revealed that this catalyst is highly effective, affording a HMF yield of 95.2% at a relatively mild temperature of 60 °C with a prolonged reaction time of 9 h in the presence of 1 mol% acid species using THF as a solvent (Supplementary Table 8 and Fig. 23). To show the practical applicability of PSS-30IL-SO₃H, dehydration of fructose was performed at a relatively large scale (2.0 g of fructose) and a HMF yield of 98.7% was obtained. Notable, HMF was isolated as the sole product after simple filtration of the catalyst and evaporation of THF (see ¹H NMR spectrum in Supplementary Fig. 24). To test the heterogeneity of PSS-30IL-SO₃H, a hot filtration and then a successive reagent addition experiment were performed. After completion of the first run, PSS-30IL-SO₃H was hot filtered and fresh starting material (fructose 100 mg) was directly added to the reaction mixture of the first run and the reaction was monitored as before. No further HMF was yielded, indicating that the catalyst is stable under the reaction conditions and does not undergo leaching, which was also reflected by the neutrality (pH = 7) of the product solution. To further validate the potential application, the recyclability of PSS-30IL-SO₃H was evaluated by monitoring the performance over five consecutive runs, revealing that both the productivity and selectivity to HMF were maintained (Supplementary Tables 9 and 10).

**Discussion**

With this success, we turned our attention to introduce catalytically active sites to other PSSs to deliver activity. They were synthesized in a similar fashion and it was found that this strategy is broadly applicable, with the synthetic details and characterizations compiled in the Supplementary Information (see Supplementary Fig. 25 and 26 and Tables 11 and 12). The sulfonic acid groups incorporated in PSS-NMP and PSS-DMSO (PSS-xNMP-SO₃H and PSS-xDMSO-SO₃H, Supplementary Fig. 27

and 28) also show exceptional catalytic performance in the dehydration of fructose to HMF in THF, affording optimized HMF yields of 93.4% and 94.1%, respectively, which rivals to the literature results performed in NMP or DMSO. The solvation effect as a result of the interaction between the densely populated hydrogen bond acceptor moieties in those catalytic systems and fructose is also detected, as evidenced by the greatly decreased crystallinity of fructose after being grinded with those polymers. Similarly, the fructose crystallinity in the resultant composites decreased along with the increased density of solvent moieties in them as described above (Supplementary Fig. 29 and 30). On the basis of experimental results aforementioned and some previous reports[14–16,56–59], a tentative mechanism is proposed for the dehydration of fructose to HMF catalyzed by PSSs-SO₃H as illustrated in Supplementary Fig. 31: these solvent moieties act as a polar aprotic solvent, functioning as a proton acceptor and forming a hydrogen-bonding network with the hydroxyl groups on fructose, increasing the stability of the furanose tautomers, and resulting in increased HMF selectivity as well as reducing humin formation relative to catalysts without these functionalities. We deduce that a high density of solvent moieties well-oriented around the acid sites could boost the synergistic effect in the confined nanospace, thus leading to their high catalytic performance in terms of both activity and selectivity. Detailed mechanistic studies to probe the intermediates during the dehydration of fructose to HMF could be necessary, and research along this line will be conducted in the near future.

To show the general applicability of this strategy in regulating the catalytic performance, a basic functionality was introduced into PSSs. Accordingly, following a similar copolymerization procedure, basic monomers (1-vinylimidazole) were successfully incorporated into PSS-IL (PSS-xIL-IM, where x refers to the mole ratio of V-IL and 1-vinylimidazole, textural parameters of these polymers are shown in Supplementary Table 13). Given the increasing interest in the transformation of glucose to HMF, one-pot cascade synthesis involving base-catalyzed isomerization of glucose to fructose and acid-catalyzed dehydration of fructose to HMF was chosen. Following a similar trend of that observed for PSS-xIL-SO₃H, with an increase of ionic moieties in the resultant polymers, both the reaction rate and selectivity are greatly improved leading to a higher fructose to HMF yield, reaching 84.3% of the combination of PSS-30IL-IM and PSS-30IL-SO₃H compared to only 19.5% of the mixture of PSS-5IL-IM and PSS-5IL-SO₃H after 3 h at 120 °C using THF as the reaction medium (Supplementary Table 14). The drastic solvation effect on the yield were further demonstrated by comparison of HMF yields

catalyzed by PSS-30IL-IM and PSS-30IL-SO$_3$H to that of the combination of PDVB-IM and PDVB-SO$_3$H, which were synthesized from the copolymerization of divinylbenzene with 1-vinylimidazole or sodium *p*-styrene sulfonate, followed by ion-exchange with HCl, respectively, at a mole ratio of 30 to 1. Under identical conditions, an HMF yield enhancement of up to 10-fold was observed in the catalysts bearing ionic moieties relative to those in the absence of those species.

Therefore, the synthetic methodology described here is a general method for the construction of a solvent-like environment around the active sites to modulate their performance, which shows great promise as alternatives to unreadily separable solvents to simplify the separation procedure. The porous polymeric analogs of solvents would provide potential platforms for the design and development of highly stable, efficiently active, and excellently recyclable heterogeneous catalysts by taking the advantages of their high surface areas, large pore volumes, hierarchical porosity, controllable solvent-like environments, adjustable active sites at the molecular level, in conjugation with their readily separable property.

In summary, we have developed a general and effective methodology for the construction of solvent-like environments to modulate the performance of active sites in heterogeneous catalysts, as demonstrated by the incorporation of sulfonic acid or imidazole groups in porous polymeric analogs of solvents such as ionic liquids, NMP, and DMSO. Due to the retained strong solvation ability of the densely populated solvent moieties to fructose, the sulfonic acid group incorporated PSSs show excellent catalytic performance in the dehydration of fructose to HMF using THF as a readily separable solvent, far outperforming those grafted on more conventional supports and corresponding homogeneous catalysts. Together with our process for the conversion of fructose to HMF as a sole product, reaching a yield that exceeds the best performance of all systems reported in the literature and even surpassing their performance in corresponding desired solvent media, this strategy thereby affords an attractive route for highly efficient transformation of biomass to a versatile platform compound. As well as being generally applicable of itself, the synthetic methodology described here provides not only ways to circumvent the challenge associated with the separation unfriendly solvent usage in organic transformations, but also important insights into the design and construction of sophisticated reaction environments to control over the performance of active species in heterogeneous catalysts.

## Methods

**Materials and measurements**. Solvents were purified according to standard laboratory methods. THF was distilled over LiAlH$_4$. DMF was distilled over CaH$_2$. Fructose was dried at 40 °C under vacuum for 24 h. 4-Bromostyrene was washed with NaOH aqueous (1 M), dried over CaH$_2$, and then distilled under vacuum. 1-Vinylimidazole, 4-vinylbenzyl chloride, and 2-pyrrolidinone were purchased from Aldrich and used as received. Amberlyst-15 (Aldrich) was washed with Milli-Q water, dried overnight at 80 °C. The purity of the synthesized compounds was confirmed by liquid NMR analysis (Supplementary Fig. 32).

The gas adsorption isotherms were collected on the surface area analyzer ASAP 2020. The N$_2$ sorption isotherms were measured at 77 K using a liquid N$_2$ bath. Scanning electron microscopy (SEM) images were collected using a Hitachi SU 1510. Transmission electron microscopy (TEM) was performed on Hitachi HT-7700 or JEOL2100F. IR spectra were recorded using a Thermo Scientific NICOLET iS5 spectrometer. High-angle-annular-dark-field (HAADF) scanning, STEM imaging, and energy dispersive X-ray spectroscopy (EDX) mapping were carried out by Titan ChemiSTEM operated at 200 kV. X-ray photoelectron spectroscopy (XPS) spectra were performed on a Thermo ESCALAB 250 with Al Kα irradiation at θ = 90° for X-ray sources, and the binding energies were calibrated using the C1s peak at 284.9 eV. The S content of the catalysts was determined by a high frequency infrared carbon-sulfur analyzer (CS-600 series, LECO, USA). $^1$H NMR spectra were recorded on a Bruker Avance-400 (400 MHz) spectrometer. Chemical shifts are expressed in ppm downfield from TMS at δ = 0 ppm, and *J* values are given in Hz. $^{13}$C (100.5 MHz) cross-polarization magic-angle spinning (CP-MAS) NMR experiments were recorded on a Varian infinity plus 400 spectrometer equipped

with a magic-angle spin probe in a 4-mm ZrO$_2$ rotor. Powder X-ray diffraction (XRD) patterns were measured with a Rigaku Ultimate VI X-ray diffractometer (40 kV, 40 mA) using CuKα (λ = 1.5406 Å) radiation. Before the $^{31}$P MAS NMR measurements for acid strength evaluation, samples were placed in a Pyrex cell equipped with a stopcock. The sample cell was outgassed at 100 °C. A CH$_2$Cl$_2$ solution containing a known amount of TMPO was added to the sample cell inside a glovebox to prevent moisture adsorption. To ensure uniform adsorption of probe molecules on the sample, the loaded sample was sonicated for 1 h and then allowed to sit at ambient temperature overnight. Removal of the CH$_2$Cl$_2$ solvent was achieved by evacuation at room temperature and then evacuation (at 323 K). Finally, the sample vessel was placed in the glovebox where the sample was transferred into a ZrO$_2$ MAS rotor (4 mm o.d.) and then sealed by a gastight Kel-F cap. Then the $^{31}$P MAS NMR spectra were recorded on a Bruker Avance 500 spectrometer and (NH$_4$)$_2$HPO$_4$ was chosen as reference with $^{31}$P chemical shift at 1.0 ppm.

**Data availability**. The authors declare that all the data supporting the findings of this study are available within the article (and Supplementary Information files), or available from the corresponding author on reasonable request.

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

## Acknowledgements

We acknowledge National Natural Science Foundation of China (21333009, 21422306, and 21720102001).

## Author contributions

Q.S. and F.S.X. conceived and designed the research. Q.S., B.A., X.M., and F.S.X. drafted the manuscript. Q.S. and S.W. synthesized the materials and carried out the catalytic evaluation. S.M. assisted with additional experiments and participated in discussion. All authors discussed the results and gave approval to the final version of the manuscript.
