## [Peer Review File · Nature Communications]

Reviewers' comments:

Reviewer #1 (Remarks to the Author):

The manuscript entitled “Porous Solid ‘Solvents’ as a Designer Platform for Efficient Biomass Conversion” by Xiao et al. reports a new approach of decorating the solvent moieties into porous materials to exceptionally enhance conversions of fructose to 5-hydroxymethylfurfural (HMF) in THF. This strategy fundamentally eliminates the concerns with separation unfriendly solvents. The catalytic performance in the dehydration of fructose to HMF behaviors were characterized well. I am pleased to recommend its publication after minor revision.

1. Obviously, hydrogen bonds in this porous materials should play an important role in the catalysis. In addition, IR and NMR spectrum (especially in situ temperature measurements) can provide more information about hydrogen bonds. So, authors could add some related in-depth discussions to explore the mechanism of reaction.
2. In title of this manuscript, authors introduce the concept of “Porous Solid Solvents” to readers, which may be easily misinterpreted that there are no liquid solvents participating catalysis. Actually, they use THF as the solvent of catalysis reaction.
3. There may have some problems about calculating BET surface area from N₂ sorption isotherms in Supplementary Table 1 and Supplementary Figure 17.

Reviewer #2 (Remarks to the Author):

1- The article is interesting since it describes the preparation of novel families of organic polymers based on monomeric units similar to solvent molecules. The objective is to generate stable polymeric networks where sulphonic groups are embedded into these polymeric like-solvent matrixes, being the acid sites surrounded by solvent mimic microenvironment that increase reaction rates and improve the selectivity of the catalytic processes. Conceptually, the article is well structured and the results are well presented and logical, the idea being a promising advance inside polymeric catalytic materials, in which intrinsic solvent effect could be incorporated into the solids avoiding separation and costly isolation processes. However, from my point of view, additional experiments and characterization studies would be necessary to extend and validate this approach to more solvents, more catalytic active sites and more reaction processes. In the current form, the article would not be publishable in Nature

Communication, suggestions are enclosed.

2- The introduction to active species into Porous Solids Solvents is carried out through the polymerization between vinyl-monomer-solvent with p-styrene sulfonate. But, what was the homogeneity achieved during this process, regarding the distribution of sulphonic groups along the polymeric network?

3- Exchange post-synthesis treatment is performed to generate sulphonic groups from sodium sulphonate moieties, using HCl solutions. But, what is the exchange level finally achieved?

4- The detection of sulphonic groups was followed from IR spectroscopy. In my opinion, it would be necessary to confirm additionally the presence, amount and strength of acid sites from additional characterization techniques, such as elemental analyses, ¹³C NMR spectroscopy and/or adsorption of probe molecules.

5- In general, what is the hydrothermal stability of these type of polymeric Porous Solid Solvents?

6- The solvent effect of the materials was analyzed introducing fructose in the reaction medium in presence of solid polymer containing 1-methyl-2-pyrrolidinone (NMP) as monomer solvent units. In this case, it is observed the rupture of the hydrogen network of fructose via coordinating the fructose hydroxyl groups to the hydrogen bond acceptor in Porous Solid Solvents like their liquid analogues. Was this same effect observed with the polymers which contain DMSO or ionic liquid into their framework?

7- Acid-base titration of the solids was carried out to evaluate the amount of sulphonic groups incorporated in the polymers. However, this type of titrations in solids are frequently inaccurate. Another methods to analyze the content of sulphonic groups would be necessary.

8- If one of the main objectives of the study is the elimination of the solvents during the catalytic reaction processes, having account that this effect would be present in the polymeric matrix, why is THF used in the catalytic tests performed in the article? What about the “solvent-free” reactions?

9- The catalytic tests were carried out with the polymers containing sulphonic groups in presence of ionic liquid fragments. But, what is the catalytic behavior of another polymers studied in the work, such as those containing DMSO or NMP units?

10- In general, in the catalytic study, it would be necessary to incorporate additional data related with leaching studies of sulphonic groups, blank reactions, influence of number of active sites on

the catalyst reactivity, estimation of initial reaction rates and TON/TOFs values, etc. to establish a correct comparison with another similar sulphonic-polymer catalysts.

11- The reaction chosen was the conversion of fructose to 5-hydroxymethylfurfural. But, another reactions could be tested to validate the concept proposed by the authors.

Reviewer #3 (Remarks to the Author):

This article reports an approach to functionalize polymer matrices with organic solvents and solid acid sites, resulting in heterogeneous catalysts which mimic liquid-phase solvents for biomass conversion reactions. Several scientific issues must be addressed before the article is suitable for publication. Please see our comments below.

1) The authors claim that the reactivity of the acidic porous solid solvents “rivals” the reactivity with the corresponding liquid-phase solvents (e.g. NMP, DMSO, ionic liquid). However, the results are in some cases quite different. The PSS materials actually display higher selectivity than their liquid-phase counterparts. For example, PSS-30IL-SO₃H has 99% selectivity at 100% conversion (table 2, entry 1), while Amberlyst-15 + EMIMBr shows only 52% selectivity at 100% conversion (Table S3, Entry 6). Similarly, PSS-30NMP-SO₃H and PSS-30DMSO-SO₃H (Table S3, Entry 4 and 5) have higher selectivities than Amberlyst-15 with NMP or DMSO (Table S3, entries 7 and 8). These results suggest that the porous solid solvents do not create an identical chemical environment to the liquid-phase solvents. The authors should explain these differences between the solid solvents and liquid-phase solvents, or at least provide a reasonable hypothesis.

2) Concerning the discussion of the effect of solvents on rates and selectivities in biomass conversion reactions, there are several key points missing which the authors should address. The authors should discuss the effect of solvents on the stability of the proton, reactant, and protonated transition state. On Page 9, the authors claim that these solvents “perturb the reaction”, but this is vague. Second, the authors should discuss the effect of solvent on product selectivity via changing the relative rates of desired reactions and undesired reactions (e.g. humin formation reactions). Please provide a more thorough discussion of the mechanistic role the solvent plays in affecting the rates of acid-catalyzed reactions.

Relevant literature in this context:

- i) M. A. Mellmer, C. Sener, J. M. R. Gallo, J. S. Luterbacher, D. M. Alonso and J. A. Dumesic, "Solvent Effects in Acid-Catalyzed Biomass Conversion Reactions," *Angewandte Chemie International Edition*, 2014, 53, 11872-11875.
- ii) M. A. Mellmer, C. Sanpitakseree, B. Demir, P. Bai, K. Ma, M. Neurock and J. A. Dumesic, "Solvent-enabled control of reactivity for liquid-phase reactions of biomass-derived compounds," *Nature Catalysis*, 2018, 1, 199-207.

iii) T. W. Walker, A. K. Chew, H. Li, B. Demir, Z. C. Zhang, G. W. Huber, R. C. Van Lehn and J. A. Dumesic, "Universal kinetic solvent effects in acid-catalyzed reactions of biomass-derived oxygenates," *Energy & Environmental Science*, 2018.

iv) Madon, R. J. & Iglesia, E. Catalytic reaction rates in thermodynamically non-ideal systems. *J. Mol. Catal. A Chem.* 163, 189–204 (2000).

v) Cox, B. G. *Acids and Bases: Solvent Effects on Acid* (Oxford Univ. Press, Oxford, 2013).

3) Based on Figure 3, the authors claim that “spatial continuity” of solvent moieties is important for catalytic activity. However, these data only display an effect of the density of solvent moieties. There is no direct evidence that these solvent moieties are spatially continuous at higher loadings. The authors should either provide evidence of this fact, or remove claims of “spatial continuity”.

4) The catalyst recycling experiments (Table 2, Entry 9) were done at 100% conversion. Therefore these experiments do not show that the catalyst is stable. These experiments should be repeated at a shorter reaction time where the conversion is <100%. If the catalyst is not stable the authors should discuss and identify the sources of deactivation.

5) The authors should test for leaching of sulfonic acid groups from the catalyst. The pH of the product solution can be checked. Alternatively, the liquid reactor effluent could be tested for catalytic activity.

6) Is the synthesis of the porous solid solvents in this manuscript a novel procedure? If this synthesis procedure builds on procedures in the literature, please cite these sources.

7) Have the authors tried their approach with water as the liquid-phase? This result should be mentioned in the manuscript or SI. As the authors argue that THF is more easily separable than some of the other replaced solvents, it is natural to consider whether water solvent is also suitable. Of course, carrying out this reaction in pure water would likely result in the decomposition of HMF to form levulinic acid and formic acid. However, a minimum amount of water is often present in liquid-phase biomass-conversion processes to facilitate solubility of the reactants.

8) Another point regarding the use of THF as solvent, as opposed to water: the authors report that HMF was quantified by GC (Supplementary Info). How was fructose conversion quantified? Furthermore, the solubility of fructose in THF is very low – this would make fructose quantification difficult.

9) Page 11 – the authors mention “Acid-base titration” but do not report the method used. Additionally, for the data in Table 2, the authors report that each experiment was run at a fixed number of acid sites – for these data it is especially important to report the method of acid site counting.

10) Table 2- the authors should clarify the meaning of the values in parentheses. Also, we believe there is a typo for Entries 6 and 7 – the superscript should be “d”, not “c”.

11) In several instances (e.g. pg 12), the authors discuss product yields, but do not note the

different conversion levels. The authors should mention both the conversion and the selectivity when comparing results.

12) Page 16, the authors mention a 95.2% yield at 60C with a reaction time of 9h. The catalyst loading should also be mentioned here, as the reaction time required depends on the catalyst loading.

13) Page 16, the authors mention “long-term stability” of their catalyst by testing the reaction at a larger scale. This experiment does not test catalyst stability.

14) There are some duplicate entries in Table S3.

15) There are several instances of imprecise language used in the article. For example, “exceptional conversions” (pg 1) and “Gratifyingly...” (pg 17). Page 6- what is meant by “solvent tests”?

We really appreciate the constructive comments and suggestions from the reviewers, and we have revised the manuscript accordingly as detailed in the responses below. The corresponding changes have been highlighted in yellow in the main text and supplementary information.

Reviewer #1:

Comment 1: The manuscript entitled “Porous Solid ‘Solvents’ as a Designer Platform for Efficient Biomass Conversion” by Xiao et al. reports a new approach of decorating the solvent moieties into porous materials to exceptionally enhance conversions of fructose to 5-hydroxymethylfurfural (HMF) in THF. This strategy fundamentally eliminates the concerns with separation unfriendly solvents. The catalytic performance in the dehydration of fructose to HMF behaviors were characterized well. I am pleased to recommend its publication after minor revision.

Response: We appreciate the high comments and support from the reviewer.

Comment 2: Obviously, hydrogen bonds in this porous materials should play an important role in the catalysis. In addition, IR and NMR spectrum (especially in situ temperature measurements) can provide more information about hydrogen bonds. So, authors could add some related in-depth discussions to explore the mechanism of reaction.

Response: We thank the reviewer for the valuable comment. Per the reviewer’s suggestion, variable temperature IR spectra of the composite of finely grinded fructose and PSSs, as exemplified by PSS-NMP, at a mass ratio of 10/1 (fructose@PSS-NMP) were collected. It was shown that at room temperature the C=O vibration shifted from 1677 cm^{-1} for pristine PSS-NMP to 1663 cm^{-1} for fructose@PSS-NMP, meanwhile the peak associated with the OH group of fructose shifted from 1044 cm^{-1} to 1054 cm^{-1} for the composite, thereby confirming that strong interactions exist between fructose and the NMP moieties in PSS-NMP. Such interactions are relatively strong as evidenced by the fact that increasing the temperature to 100 °C, negligible changes were observed. Based on these experimental results and other published references, we have now included some discussions associated with the mechanistic role of a solvent on affecting the performance of acid sites in the dehydration of fructose to form HMF. We have also included a figure of this mechanism in the Supplementary Information as Figure 30.

Comment 3: In title of this manuscript, authors introduce the concept of “Porous Solid Solvents” to readers, which may be easily misinterpreted that there are no liquid solvents participating catalysis. Actually, they use THF as the solvent of catalysis reaction.

Response: We appreciate the reviewer for pointing this out. To avoid misinterpretation, we have revised the title into “Creating Solvation Environments in Heterogeneous Catalysts for Efficient Biomass Conversion”.

Comment 4: There may have some problems about calculating BET surface area from N₂ sorption isotherms in Supplementary Table 1 and Supplementary Figure 17.

Response: We thank the reviewer for the comment. We have proofread the submitted data with the original ones and they are correct. However, to further confirm the accuracy of these results, we tested newly prepared materials. In comparison with the former samples, their differences in BET surface areas are in the range of $\pm 5\%$.

The reviewer may be curious as to why the surface areas of the resultant polymers do not decrease along with the increment of sodium *p*-styrene sulfonate incorporated? This is probably due to the incorporation of sodium *p*-styrene sulfonate leading to a varied arrangement of the building blocks and thereby the pore structure, which is verified by the shape of the N₂ sorption isotherms.

Reviewer #2:

Comment 1: The article is interesting since it describes the preparation of novel families of organic polymers based on monomeric units similar to solvent molecules. The objective is to generate stable polymeric networks where sulphonic groups are embedded into these polymeric like-solvent matrixes, being the acid sites surrounded by solvent mimic microenvironment that increase reaction rates and improve the selectivity of the catalytic processes. Conceptually, the article is well structured and the results are well presented and logical, the idea being a promising advance inside polymeric catalytic materials, in which intrinsic solvent effect could be incorporated into the solids avoiding separation and costly isolation processes. However, from my point of view, additional experiments and characterization studies would be necessary to extend and validate this approach to more solvents, more catalytic active sites and more reaction processes. In the current form, the article would not be publishable in Nature Communication, suggestions are enclosed.

Response: We appreciate the reviewer for taking the time to evaluate our manuscript and providing constructive comments. The concerns raised by the reviewer have been responded point-by-point as listed below.

Comment 2: The introduction to active species into Porous Solids Solvents is carried out through the polymerization between vinyl-monomer-solvent with *p*-styrene sulfonate. But, what was the homogeneity achieved during this process, regarding the distribution of sulphonic groups along the polymeric network?

Response: We thank the reviewer for the comment. To determine the distribution of sulfonate groups in the resultant catalysts, high-resolution TEM-EDX elemental mapping was carried out.

It is shown that sulfur species, a signature of the sulfonate groups, is indeed located throughout the samples, as exemplified by PSS- x IL-SO₃H ($x = 30$ and 5), thereby providing evidence that the sulfonate groups are homogeneously incorporated in the PSSs.

Comment 3: Exchange post-synthesis treatment is performed to generate sulphonic groups from sodium sulphonate moieties, using HCl solutions. But, what is the exchange level finally achieved?

Response: We are thankful for the reviewer's comment. To evaluate the exchange level between Na⁺ and H⁺ ions, high-resolution TEM-EDX elemental mapping and XPS analyses were performed as represented by PSS-30IL-SO₃Na/H and PSS-5IL-SO₃Na/H. These results show that negligible Na species were detected in both PSS-30IL-SO₃H and PSS-5IL-SO₃H, indicative of the high level of ion exchange.

Comment 4: The detection of sulphonic groups was followed from IR spectroscopy. In my opinion, it would be necessary to confirm additionally the presence, amount and strength of acid sites from additional characterization techniques, such as elemental analyses, ¹³C NMR spectroscopy and/or adsorption of probe molecules.

Response: We appreciate the reviewer for the valuable suggestions. To confirm the successful incorporation of sulfonic groups, in addition to IR spectra, XPS and high-resolution TEM-EDX elemental mapping analyses were carried out, verifying the presence of S species in the resultant copolymers. Furthermore, to determine the content of the incorporated sulfonic groups, infrared absorption carbon-sulfur analysis was carried out, revealing that the incorporated S content was consistent with the theoretical values as well as the amounts obtained from acid-base titration. Moreover, to evaluate the acid strength of sulfonic groups in the resultant samples, studies associated with the adsorption of a probe molecule were carried out. Given the sensitivity and isotropicity of ³¹P chemical shift of the probe trimethylphosphine oxide (TMPO) according to its interaction strength with a Brønsted acid site, it has proven to be an informative tool for identifying the acidity of multiple acid sites and thereby is our choice. ³¹P MAS NMR spectra of TMPO after interaction with PSS-30IL-SO₃H and PSS-5IL-SO₃H samples show a singlet peak at 64.8 and 65.3 ppm, respectively, appearing to possess a moderate strength. The narrow NMR signal further suggests the homogeneity of the acid sites in these samples. We have updated these results in the revised manuscript.

Comment 5: In general, what is the hydrothermal stability of these type of polymeric Porous Solid Solvents?

Response: We thank the reviewer for the comment. To evaluate the hydrothermal stability of the PSSs, boiling water treatment was carried out. After one week of treatment, the PSSs still displayed high porosities and negligible decreases in surface area and pore volume were observed in comparison with the pristine ones, verifying their excellent hydrothermal stability. We have summarized these textural parameters in the Supplementary Information as Table 1.

Comment 6: The solvent effect of the materials was analyzed introducing fructose in the reaction medium in presence of solid polymer containing 1-methyl-2-pyrrolidinone (NMP) as monomer solvent units. In this case, it is observed the rupture of the hydrogen network of fructose via coordinating the fructose hydroxyl groups to the hydrogen bond acceptor in Porous Solid Solvents like their liquid analogues. Was this same effect observed with the polymers which contain DMSO or ionic liquid into their framework?

Response: We thank the reviewer for the comment. Similar studies have been done with respect to PSS-DMSO and PSS-IL, however, due to the overlap of multi-signals in those IR spectra, we didn't obtain any informative results.

Comment 7: Acid-base titration of the solids was carried out to evaluate the amount of sulphonic groups incorporated in the polymers. However, this type of titrations in solids are frequently inaccurate. Another methods to analyze the content of sulphonic groups would be necessary.

Response: We appreciate the reviewer for the criticism. To validate the acid-base titration results, infrared absorption carbon-sulfur analyses were carried out to evaluate the S species content in the resultant samples, revealing that they are in good agreement with each other. It is worthy to mention that the experimental S content is consistent with the theoretical values, indicating that $-SO_3H$ groups are virtually quantitatively incorporated into the resultant polymers. We have summarized the content of sulfonic groups in the resultant polymers derived from various analyses in the Supplementary Information as Table 2.

Comment 8: If one of the main objectives of the study is the elimination of the solvents during the catalytic reaction processes, having account that this effect would be present in the polymeric matrix, why is THF used in the catalytic tests performed in the article? What about the "solvent-free" reactions?

Response: We are thankful to the reviewer for the insightful criticisms. We have tested the reactions under "solvent-free" conditions as exemplified by mixing PSS-30IL- SO_3H (100 mg) and fructose (100 mg) and then heating up to 120 °C. The yield of 5-hydroxymethylfurfural (HMF) as a function of reaction time indicated that it was always below 7% within 2 h, at which time fructose reached full conversion (Supplementary Fig. 23). It is noteworthy that the color of the reaction mixture turned from light yellow to dark brown, indicative of the occurrence of side reactions, probably due to the instability of HMF under solvent-free and heating conditions, which also has been noted in the reference (*Angew. Chem. Int. Ed.* 2016, 55, 8838); while, in the presence of THF, the yield of HMF can reach 98.8%, indicating that solvent is still needed for mass transfer to bring the reaction under control, at least in this case.

However, in addition to these, there is another important role of solvents. They offer a suitable solvation environment to exert non-covalent interactions to provide control over the reactions. Given the extreme complicacy in separation of the products from the solvents, such

as DMSO, NMP, and ionic liquids, the focus of this work is to develop alternatives that could mimic the solvation environment of these solvents to aid the transformations where those types of solvent are preferred to achieve high yields. Our strategy to fulfill this task is to construct the desired solvent moieties into porous materials, which in turn serve as platforms for introducing catalytic species and to perform the reaction using a separate-friendly solvent instead, for mass and heat transfer.

Given the fact that liquid solvent is still needed for achieving high product yield in our proof-of-concept study, to avoid confusion, we have now changed the manuscript title into “Creating Solvation Environments in Heterogeneous Catalysts for Efficient Biomass Conversion”.

Comment 9: The catalytic tests were carried out with the polymers containing sulphonic groups in presence of ionic liquid fragments. But, what is the catalytic behavior of another polymers studied in the work, such as those containing DMSO or NMP units?

Response: We appreciate the reviewer for the comment. The catalytic behavior of sulfonic groups incorporated PSS-DMSO and PSS-NMP (PSS- x DMSO-SO₃H and PSS- x NMP-SO₃H) has been systematically studied. They also show exceptional catalytic performance in the dehydration of fructose to HMF in THF, affording optimized HMF yields of 93.4% and 94.1%, respectively, which rival to the literature results performed using NMP or DMSO as a reaction medium and the details are shown in the Supplementary Information as Fig. 27 and 28.

Comment 10: In general, in the catalytic study, it would be necessary to incorporate additional data related with leaching studies of sulphonic groups, blank reactions, influence of number of active sites on the catalyst reactivity, estimation of initial reaction rates and TON/TOFs values, etc. to establish a correct comparison with another similar sulphonic-polymer catalysts.

Response: We thank the reviewer for the valuable suggestions. To show the necessity of Brønsted acid sites for the accomplishment of the reactions, control experiments were carried out. No HMF was detected in the absent of catalyst or only PSS-IL involved. To evaluate the leaching of sulfonic groups during the reaction, a hot filtration and then a successive addition experiment were performed. After completion of the first run, PSS-30IL-SO₃H was hot filtered and a fresh starting material (fructose 100 mg) was directly added to the reaction mixture of the first run and the reaction was monitored as before. No additional HMF was yielded, indicating that the catalyst is stable under the reaction conditions and does not undergo leaching, which is also verified by its excellent recyclability.

The influence of active site numbers on the catalyst reactivity was studied and catalysts with different concentrations of acid sites were achieved by co-polymerization of V-IL and sodium *p*-styrene sulfonate at various ratios (PSS- x IL-SO₃Na), followed by ion-exchange with HCl (PSS- x IL-SO₃H, $x = 5 \sim 30$), where x represents the molar ratio of V-IL and sodium *p*-styrene sulfonate. Upon an increase of the molar ratio of V-IL with sulfonic acid, augmented productivity as well as selectivity to HMF was observed in the presence of the same sulfonic

acid content. In line with its maximum ionic component ratio among the samples tested, PSS-30IL-SO₃H exhibited the highest HMF yield (98.8%). As the mole ratio decreased from 30 to 5, the HMF yield decreased substantially from 98.8% to 13.6%. These results imply the importance of the density of the ionic component on the catalytic performance of the resultant catalysts. Very similar trends were also observed in the sulfonic acid groups incorporated PSS-NMP and PSS-DMSO (PSS-xNMP-SO₃H and PSS-xDMSO-SO₃H). In brief, in our testing range, lowering the number of sulfonic acid sites to maintain a high density of solvent moieties in the resultant catalysts favored the reaction outcome.

Furthermore, per the reviewer's suggestion, the initial reaction rates and TON/TOF values were estimated to establish a correct comparison with other catalytic systems. To keep the fructose conversion below 15% and thereby ensuring the accuracy of TOF values calculation, catalytic reactions were performed under a relatively high substrate to catalyst ratio (S/C = 400) in the presence of THF at 120 °C. PSS-xIL-SO₃H (x = 30, 20, 15, 10, 5) afforded an initial reaction rate of 0.064, 0.048, 0.042, 0.032, and 0.0095 mM min⁻¹, respectively, at the beginning 5 min. We have updated these results in the revised manuscript.

Comment 11: The reaction chosen was the conversion of fructose to 5-hydroxymethylfurfural. But, another reactions could be tested to validate the concept proposed by the authors.

Response: We appreciate the reviewer for the constructive comment. To show the general applicability of the strategy demonstrated herein in regulating the catalytic performance, a basic functionality was introduced into PSSs. Accordingly, following a similar copolymerization procedure, basic monomers (1-vinylimidazole) were successfully incorporated into PSS-IL (PSS-xIL-IM, where x refers to the mole ratio of V-IL and 1-vinylimidazole, textural parameters of these polymers are shown in Supplementary Table 13). Given the increasing interest in the transformation of glucose to HMF, one-pot cascade synthesis involving base-catalyzed isomerization of glucose to fructose and acid-catalyzed dehydration of fructose to HMF was chosen. Following a similar trend of that observed for PSS-xIL-SO₃H catalyzed fructose to HMF, with an increase of ionic moieties in the resultant polymers, a greatly improved fructose to HMF yield were obtained, reaching 84.3% of the combination of PSS-30IL-IM and PSS-30IL-SO₃H compared to only 19.5% of the mixture of PSS-5IL-IM and PSS-5IL-SO₃H after 3 h at 120 °C using THF as the reaction medium (Supplementary Table 14). The drastic solvation effect on the yield was further demonstrated by comparison of HMF yield catalyzed by PSS-30IL-IM and PSS-30IL-SO₃H to that of the combination of PDVB-IM and PDVB-SO₃H, synthesized from the co-polymerization of divinylbenzene with 1-vinylimidazole or sodium *p*-styrene sulfonate, followed by ion-exchange with HCl, at a mole ratio of 30 to 1, respectively. Under identical conditions, an HMF yield enhancement of up to 10-fold was observed in the catalysts bearing ionic moieties relative to those in the absence of those species.

Reviewer #3:

Comment 1: This article reports an approach to functionalize polymer matrices with organic solvents and solid acid sites, resulting in heterogeneous catalysts which mimic liquid-phase solvents for biomass conversion reactions. Several scientific issues must be addressed before the article is suitable for publication. Please see our comments below.

Response: We appreciate the reviewer for taking the time to evaluate our manuscript and providing constructive comments. The concerns raised by the reviewer have been responded point-by-point as listed below.

Comment 2: The authors claim that the reactivity of the acidic porous solid solvents “rivals” the reactivity with the corresponding liquid-phase solvents (e.g. NMP, DMSO, ionic liquid). However, the results are in some cases quite different. The PSS materials actually display higher selectivity than their liquid-phase counterparts. For example, PSS-30IL-SO₃H has 99% selectivity at 100% conversion (table 2, entry 1), while Amberlyst-15 + EMIMBr shows only 52% selectivity at 100% conversion (Table S3, Entry 6). Similarly, PSS-30NMP-SO₃H and PSS-30DMSO-SO₃H (Table S3, Entry 4 and 5) have higher selectivities than Amberlyst-15 with NMP or DMSO (Table S3, entries 7 and 8). These results suggest that the porous solid solvents do not create an identical chemical environment to the liquid-phase solvents. The authors should explain these differences between the solid solvents and liquid-phase solvents, or at least provide a reasonable hypothesis.

Response: We thank the reviewer for the insightful comment. We ascribe the disparity in selectivity of HMF catalyzed by PSS-30IL-SO₃H or Amberlyst-15 with corresponding solvent to their difference in acid strength. The ³¹P MAS NMR spectra of trimethylphosphine oxide (TMPO) after interaction with PSS-30IL-SO₃H and Amberlyst-15 samples shows a singlet peak at 64.8 and 82.5 ppm, respectively, indicating that the TMPO molecules exhibit higher interaction strength with Brønsted acid sites in Amberlyst-15 than those in PSS-30IL-SO₃H (Supplementary Fig. 21). It is well documented that large amounts of strong acid sites may lead to side reactions and thus reduce the yield of HMF (ChemSusChem 2017, 10, 1669-1674; Green Chem. 2013, 15, 3367-3376; Chem. Commun. 2012, 48, 5850-5852). To rationalize our assumption, a porous polymer-based catalyst (PDVB-SO₃H) with similar acid strength to that of PSS-30IL-SO₃H was used for comparison. This material was synthesized by co-polymerization of divinylbenzene and sodium *p*-styrene sulfonate with a mole ratio of 30/1, followed by ion-exchange with 1M HCl. Significant improvements were observed when the reactions were operated in EMIMBr, DMSO, or NMP in the presence of PDVB-SO₃H compared to Amberlyst-15. However, using THF as the reaction medium, a very low HMF selectivity was offered by PDVB-SO₃H (Supplementary Table 5), confirming the importance of the desired solvation environment around the active sites to achieve high performance.

Comment 3: Concerning the discussion of the effect of solvents on rates and selectivities in biomass conversion reactions, there are several key points missing which the authors should address. The authors should discuss the effect of solvents on the stability of the proton, reactant, and protonated transition state. On Page 9, the authors claim that these solvents “perturb the reaction”, but this is vague. Second, the authors should discuss the effect of solvent on product selectivity via changing the relative rates of desired reactions and undesired reactions (e.g. humin formation reactions). Please provide a more thorough discussion of the mechanistic role the solvent plays in affecting the rates of acid-catalyzed reactions.

Relevant literature in this context:

- i) M. A. Mellmer, C. Sener, J. M. R. Gallo, J. S. Luterbacher, D. M. Alonso and J. A. Dumesic, "Solvent Effects in Acid-Catalyzed Biomass Conversion Reactions," *Angewandte Chemie International Edition*, 2014, 53, 11872-11875.
- ii) M. A. Mellmer, C. Sanpitakseree, B. Demir, P. Bai, K. Ma, M. Neurock and J. A. Dumesic, "Solvent-enabled control of reactivity for liquid-phase reactions of biomass-derived compounds," *Nature Catalysis*, 2018, 1, 199-207.
- iii) T. W. Walker, A. K. Chew, H. Li, B. Demir, Z. C. Zhang, G. W. Huber, R. C. Van Lehn and J. A. Dumesic, "Universal kinetic solvent effects in acid-catalyzed reactions of biomass-derived oxygenates," *Energy & Environmental Science*, 2018.
- iv) Madon, R. J. & Iglesia, E. Catalytic reaction rates in thermodynamically non-ideal systems. *J. Mol. Catal. A Chem.* 163, 189-204 (2000).
- v) Cox, B. G. *Acids and Bases: Solvent Effects on Acid* (Oxford Univ. Press, Oxford, 2013).

Response: We are thankful to the reviewer for the constructive suggestions. Based on these important references and other published results, some discussions associated with the mechanistic role of a solvent on affecting the performance of acid sites in the dehydration of fructose to form HMF have been proposed, and the aforementioned references have also been properly cited.

Comment 4: Based on Figure 3, the authors claim that “spatial continuity” of solvent moieties is important for catalytic activity. However, these data only display an effect of the density of solvent moieties. There is no direct evidence that these solvent moieties are spatially continuous at higher loadings. The authors should either provide evidence of this fact, or remove claims of “spatial continuity”.

Response: We appreciate the reviewer for the comment. Given the 3D pore structure of the polymers studied, we reasoned that a high concentration of solvent moieties introduced would result in a better spatial continuity of those species therein. However, to avoid confusion, we have revised the claim of “spatial continuity” into “high density”.

Comment 5: The catalyst recycling experiments (Table 2, Entry 9) were done at 100% conversion. Therefore these experiments do not show that the catalyst is stable. These experiments should be

repeated at a shorter reaction time where the conversion is <100%. If the catalyst is not stable the authors should discuss and identify the sources of deactivation.

Response: We thank the reviewer for the valuable criticism. Per the reviewer's suggestion, we evaluated the catalyst recyclability when the yield of HMF reached to around 60%. It is shown that PSS-30IL-SO₃H can maintain its performance in terms of both activity and selectivity for more than 5 cycles, thereby indicative of its excellent stability (see Supplementary Table 10).

Comment 6: The authors should test for leaching of sulfonic acid groups from the catalyst. The pH of the product solution can be checked. Alternatively, the liquid reactor effluent could be tested for catalytic activity.

Response: We appreciate the reviewer for the comments. To test the heterogeneity of PSS-30IL-SO₃H, a hot filtration and then a successive reagent addition experiment were performed. After completion of the first run, PSS-30IL-SO₃H was hot filtered and a fresh starting material (fructose 100 mg) was directly added to the reaction mixture of the first run and the reaction was monitored as before. No further HMF was yielded, indicating that the catalyst is stable under the reaction conditions and does not undergo leaching, which is also verified by its excellent recyclability as demonstrated above. In addition, per the reviewer's suggestion, the pH of the product solution has also been evaluated and it was neutral.

Given the low solubility of fructose in THF, using a liquid reactor effluent strategy to test the long-term stability of the catalyst is not applicable in this case.

Comment 7: Is the synthesis of the porous solid solvents in this manuscript a novel procedure? If this synthesis procedure builds on procedures in the literature, please cite these sources.

Response: We thank the reviewer for pointing this out. The strategy for synthesizing porous polymers from polymerization of vinyl-functionalized monomers initiated by a free radical has been well established and the representative references have been properly cited in the revised manuscript. Considering its high efficiency and general applicability, we employed the similar polymerization reaction in this work for the preparation of new types of porous polymer materials.

Comment 8: Have the authors tried their approach with water as the liquid-phase? This result should be mentioned in the manuscript or SI. As the authors argue that THF is more easily separable than some of the other replaced solvents, it is natural to consider whether water solvent is also suitable. Of course, carrying out this reaction in pure water would likely result in the decomposition of HMF to form levulinic acid and formic acid. However, a minimum amount of water is often present in liquid-phase biomass-conversion processes to facilitate solubility of the reactants.

Response: We thank the reviewer for the comment. Per the reviewer's request, the impact of water on the HMF yield catalyzed by PSS-30IL-SO₃H was investigated. When using water as a reaction medium, a HMF yield of 7.2% was obtained after 3 h at 120 °C (fructose reached 100% conversion at this time), which is much inferior to the results using THF as a solvent, affording

a 98.8% HMF yield within 10 min. Also, the introduction of H₂O (5% or 10% volume ratio to THF) has a negative impact on the catalytic performance of PSS-30IL-SO₃H in terms of both activity and selectivity, see details in Supplementary Table 8. This is probably because the amount of solvent moieties on the PSSs for creating solvation environments is far less than when they are used as a solvent and therefore can't form the water-enriched local solvent domain as proposed in the references given by the reviewer.

Comment 9: Another point regarding the use of THF as solvent, as opposed to water: the authors report that HMF was quantified by GC (Supplementary Info). How was fructose conversion quantified? Furthermore, the solubility of fructose in THF is very low – this would make fructose quantification difficult.

Response: We thank the reviewer for the criticism. We sincerely apologize for missing the procedures of fructose quantification. To determine the conversion of fructose at a certain time, another separate reaction was carried out. After the reaction, the catalyst was separated by centrifugation and washed with water. The concentration of fructose in the combined solution was analyzed by liquid chromatography. We have included the detailed procedures in the Supplementary Information.

Comment 10: Page 11 – the authors mention “Acid-base titration” but do not report the method used. Additionally, for the data in Table 2, the authors report that each experiment was run at a fixed number of acid sites – for these data it is especially important to report the method of acid site counting.

Response: We appreciate the reviewer for the comment. In the revised Supplementary Information the detailed procedures of “Acid-base titration” have been included. In addition, to validate the acid-base titration results, infrared absorption carbon-sulfur analysis was carried out, revealing that they are in good agreement with each other. It is worthy to mention that the experimental S content was consistent with the theoretical values, indicating that -SO₃H groups are almost quantitatively incorporated into the resultant polymers. We have summarized the content of sulfonic groups in the resultant polymers derived from various analyses in the Supplementary Information as Table 2.

Comment 11: Table 2- the authors should clarify the meaning of the values in parentheses. Also, we believe there is a typo for Entries 6 and 7 – the superscript should be “d”, not “c”.

Response: We thank the reviewer for pointing this out. In the revised manuscript we have clarified the meaning of the values in parentheses and these typos in the table have also been corrected.

Comment 12: In several instances (e.g. pg 12), the authors discuss product yields, but do not note the different conversion levels. The authors should mention both the conversion and the selectivity when comparing results.

Response: We thank the reviewer for pointing this out. In the revised manuscript, both the conversion and the selectivity were mentioned when comparing the yield.

Comment 13: Page 16, the authors mention a 95.2% yield at 60 °C with a reaction time of 9 h. The catalyst loading should also be mentioned here, as the reaction time required depends on the catalyst loading.

Response: We thank the reviewer for pointing this out. The catalyst loading has been included.

Comment 14: Page 16, the authors mention “long-term stability” of their catalyst by testing the reaction at a larger scale. This experiment does not test catalyst stability.

Response: We appreciate the reviewer for the comment. We have rephrased these sentences.

Comment 15: There are some duplicate entries in Table S3.

Response: We thank the reviewer for calling attention to the error. We have revised Table S3 accordingly.

Comment 16: There are several instances of imprecise language used in the article. For example, “exceptional conversions” (pg 1) and “Gratifyingly...” (pg 17). Page 6- what is meant by “solvent tests”?

Response: We thank the reviewer for the comment. We have reworded these instances of imprecise language and deleted “solvent tests”. We also asked a native English speaker to polish the language of the manuscript thoroughly to avoid misinterpretation.

Again we thank the reviewers for the constructive comments and suggestions, which have made our manuscript much improved.

Reviewers' Comments:

Reviewer #1 (Remarks to the Author):

The manuscript has been revised well and I am pleased to recommend its acceptance.

Reviewer #3 (Remarks to the Author):

We thank for authors for their efforts in thoroughly addressing our concerns with the original manuscript. We feel that the manuscript is of much higher quality as a result, and is nearly ready for publication in Nature Communications. We still have a few minor comments. Specifically, we respectfully ask the authors to incorporate the following changes:

-Abstract, Line 15: The authors mention “excellent spatial continuity” of solvent moieties, but in the previous revision the authors agreed that it would more appropriate to state “high density”.

-Abstract, Line 18: “high conversion” should be changed to “high yields”

- Pg 5, Line 95: “which was developed by our group” is confusing, as it is unclear which method is being referred to.

-Pg 9, Line 158: “Indeed, theoretical calculations and experimental evidence demonstrate that in a great number of carbohydrate involved transformations some key intermediates identified in the reaction pathway are a complex that incorporates with those solvents to stabilize them, thus favoring the subsequent desired transformation starting from such form” is confusing and should be re-worded.

-Pg 18, line 334, “neutrality” should be clarified as referring to pH.

-Page 19 (Line 342) – “To our delight” seems subjective and should be removed.

-Page 20: Line 372 – please clarify whether the higher yield is due to a higher rate or due to a higher selectivity. In other words, please report the conversion in these two experiments.

-Supplementary Tables 10 and 14- please report the reactant conversion.

Reviewer #1

Comment 1: The manuscript has been revised well and I am pleased to recommend its acceptance.

Response: We are grateful to the reviewer for taking time to evaluate our work and support from the reviewer.

Reviewer #3

Comment 1: We thank for authors for their efforts in thoroughly addressing our concerns with the original manuscript. We feel that the manuscript is of much higher quality as a result, and is nearly ready for publication in Nature Communications. We still have a few minor comments. Specifically, we respectfully ask the authors to incorporate the following changes:

Response: We appreciate the reviewer for taking the time to evaluate our manuscript again and providing constructive comments. The concerns raised by the reviewer have been responded point-by-point as listed below.

Comment 2: Abstract, Line 15: The authors mention “excellent spatial continuity” of solvent moieties, but in the previous revision the authors agreed that it would more appropriate to state “high density”.

Response: We thank the reviewer for pointing this out. We have revised “excellent spatial continuity” into “high density”.

Comment 3: Abstract, Line 18: “high conversion” should be changed to “high yields”.

Response: We thank the reviewer for the comment. We have changed “high conversion” into “high yields”.

Comment 4: Pg 5, Line 95: “which was developed by our group” is confusing, as it is unclear which method is being referred to.

Response: We appreciate the reviewer for the comment. To avoid confusion, we have specified it.

Comment 5: Pg 9, Line 158: “Indeed, theoretical calculations and experimental evidence demonstrate that in a great number of carbohydrate involved transformations some key intermediates identified in the reaction pathway are a complex that incorporates with those solvents to stabilize them, thus favoring the subsequent desired transformation starting from such form” is confusing and should be re-worded.

Response: We thank the reviewer for pointing this out. We have re-worded this sentence.

Comment 6: Pg 18, line 334, “neutrality” should be clarified as referring to pH.

Response: We thank the reviewer for the comment. We have specified “neutrality” as pH = 7.

Comment 7: Page 19 (Line 342) – “To our delight” seems subjective and should be removed.

Response: We appreciate the reviewer for pointing this out. We have deleted “To our delight” and reworded this sentence accordingly.

Comment 8: Page 20: Line 372 – please clarify whether the higher yield is due to a higher rate or due to a higher selectivity. In other words, please report the conversion in these two experiments.

Response: We thank the reviewer for the comment. We have clarified the reason for the observed high yield.

Comment 9: Supplementary Tables 10 and 14- please report the reactant conversion.

Response: We thank the reviewer for the comment. The reactant conversion has been included in supplementary Tables 10 and 14.

Again we thank the reviewers for the constructive comments and suggestions, which have made our manuscript much improved.